# Spatially resolved RNA-sequencing of the embryonic heart identifies a role for Wnt/β-catenin signaling in autonomic control of heart rate

**Silja Barbara Burkhard[1,2], Jeroen Bakkers[1,2,3]\***

[1]Hubrecht Institute-KNAW, Utrecht, Netherlands; [2]University Medical Center Utrecht, Utrecht, Netherlands; [3]Department of Medical Physiology, Division of Heart and Lungs, University Medical Center Utrecht, Utrecht, Netherlands

**Abstract** Development of specialized cells and structures in the heart is regulated by spatially - restricted molecular pathways. Disruptions in these pathways can cause severe congenital cardiac malformations or functional defects. To better understand these pathways and how they regulate cardiac development we used tomo-seq, combining high-throughput RNA-sequencing with tissue-sectioning, to establish a genome-wide expression dataset with high spatial resolution for the developing zebrafish heart. Analysis of the dataset revealed over 1100 genes differentially expressed in sub-compartments. Pacemaker cells in the sinoatrial region induce heart contractions, but little is known about the mechanisms underlying their development. Using our transcriptome map, we identified spatially restricted Wnt/β-catenin signaling activity in pacemaker cells, which was controlled by Islet-1 activity. Moreover, Wnt/β-catenin signaling controls heart rate by regulating pacemaker cellular response to parasympathetic stimuli. Thus, this high-resolution transcriptome map incorporating all cell types in the embryonic heart can expose spatially restricted molecular pathways critical for specific cardiac functions.
DOI: https://doi.org/10.7554/eLife.31515.001

**\*For correspondence:**
j.bakkers@hubrecht.eu

**Competing interests:** The authors declare that no competing interests exist.

## Introduction

The vertebrate heart exerts regular contractions to circulate nutrients and oxygen. Cardiomyocyte contraction is caused by an action potential generated by cardiac pacemaker cells: spontaneous, rhythmic membrane depolarization of the pacemaker cells enables them to trigger the neighboring working myocardium to contract. Heart disease, congenital malformations, aging, or somatic gene defects may cause pacemaker tissue dysfunction, resulting in severely disabling and potentially lethal bradycardia (inappropriately low heart rates) (*Dobrzynski et al., 2007*; *Wolf and Berul, 2006*). These conditions can be treated by implantation of an electronic pacemaker. However, this is not ideal as electronic pacemakers can cause cardiac complications such as infections, recurring need for surgery to replace malfunctioning parts and the lack of autonomic responsiveness. Furthermore, they are not ideal for pediatric patients due to the continuous need to adapt the implanted device to the growing heart. Biological pacemakers, generated in- or ex vivo are thus being explored as alternatives (*Rosen et al., 2011*; *Protze et al., 2017*). This, however, requires exquisite knowledge of the regulatory network driving pacemaker cell differentiation and function.

Although pacemaker cells are situated within, and coupled to, the surrounding cardiomyocytes, they retain a primitive myocardial identity (*Bakker et al., 2010*). This is established during cardiac development, when a myocardial progenitor cell population (Shox2[+]/Tbx18[+]/Isl1[+]/Nkx2.5[-]) becomes spatially restricted to the region connecting the sinus venosus to the atria (sinoatrial region). These

progenitor cells differentiate into Isl1$^+$/Hcn4$^+$ pacemaker cells, strictly separated from the Nkx2.5$^+$ working cardiomyocytes (*Ye et al., 2015*; *Mommersteeg et al., 2007*; *Wiese et al., 2009*). The pacemaker cells are prevented from differentiating into working myocardium by Shox2, which inhibits Nkx2.5 expression (*Espinoza-Lewis et al., 2011*; *Ye et al., 2015*; *Sun et al., 2015*). Isl1 plays a conserved role in pacemaker cells since zebrafish and mouse embryos lacking Isl1 have severe bradycardia progressing to complete loss of pacemaker function (*de Pater et al., 2009*; *Liang et al., 2015*; *Tessadori et al., 2012*). The etiology of the pacemaker phenotype is unclear and mechanisms acting downstream of Isl1 to control heart rate remain to be determined.

The cardiac pacemaker generates rhythmic action potentials through the combined and oscillating activity of ion channels on the surface membrane and calcium channels on the sarcoplasmic reticulum (*Lakatta et al., 2010*). Heart rate can vary greatly in response to physiological demand, which is regulated by the autonomic nervous system (*Gordan et al., 2015*). In the adult zebrafish heart, the receptors enabling autonomic regulation are expressed in the Isl1$^+$ pacemaker cells (*Stoyek et al., 2016*). Developing biological pacemakers that can be controlled by the autonomous nervous system will be a significant improvement from electronic pacemakers used today. Therefore, it is important to understand how autonomous control is established in pacemaker cells, of which there is very little knowledge.

Identifying novel molecular pathways that regulate the development and specialized functions of individual organs or tissues in vivo would be greatly facilitated by genome-wide transcriptome datasets with detailed spatial resolution during development. Several techniques have recently been developed that can resolve the expression of many genes without disruption of tissue organization. Fluorescent in situ sequencing (FISSEQ) is an in-situ sequencing technique based on sequencing on a solid substrate using fluorescent-tagged random hexamers. When combined with confocal imaging, FISSEQ can be applied to tissue sections or whole embryos (*Lee et al., 2014*). *Tomo*-seq combines standard RNA-sequencing using barcoded primers with histological tissue dissection and has been used to establish genome-wide transcriptome datasets with high spatial resolution of the whole zebrafish embryo and the injured adult zebrafish heart (*Junker et al., 2014*; *Kruse et al., 2016*; *Wu et al., 2016*). The advantages of the FISSEQ and *tomo*-seq techniques compared to other mRNA-seq related techniques are that: (1) there is no need for prior anatomical annotation of the tissue, (2) it does not rely on tissue dissociation and cell sorting that can influence gene expression, (3) spatial information of the transcripts within the tissue is maintained.

Here, we used one of these techniques, *tomo*-seq, to obtain the first genome-wide transcriptome map with high spatial resolution of the developing zebrafish heart. Using *tomo*-seq data, we identified sub-compartment specific gene expression signatures for the ventricle, atrium, AV canal, and the sinoatrial region, where pacemaker cells reside. We found that several members of the Wnt/β-catenin signaling pathway were differentially expressed in the sinoatrial region, resulting in active Wnt/β-catenin signaling in developing pacemaker cells. Functional analysis demonstrated that Wnt/β-catenin signaling acts downstream of Isl1 to establish parasympathetic control of heart rate. Together these results reveal a genetic pathway regulating autonomic control of pacemaker activity.

## Results

### A genome-wide transcriptome dataset with high spatial resolution of the developing zebrafish heart

At 2 days post-fertilization (2 dpf), the zebrafish heart has developed into a looped structure with a recognizable atrium and ventricle that is able to sustain blood circulation in the larvae. At this stage, functional Isl$^+$ pacemaker cells are located in the sinoatrial region (*Arrenberg et al., 2010*; *Tessadori et al., 2012*). To obtain a transcriptome map of the developing heart at that stage with spatial resolution, we applied the *tomo*-seq method on dissected hearts from 2-day-old zebrafish embryos (*Figure 1A*). We generated independent datasets from three wild-type hearts. We chose the most robust dataset (heart#1; for details see Materials and methods section) for further analysis and validation. In short, the isolated cryo-preserved hearts were sectioned into forty 10 μm sections along the anterior-posterior axis. Total RNA was isolated from each section, barcoded and processed for mRNA-sequencing (*Hashimshony et al., 2016*). We obtained a gene expression dataset with expression information for ~13,000 genes (*Figure 1—figure supplement 1* and *Figure 1—*

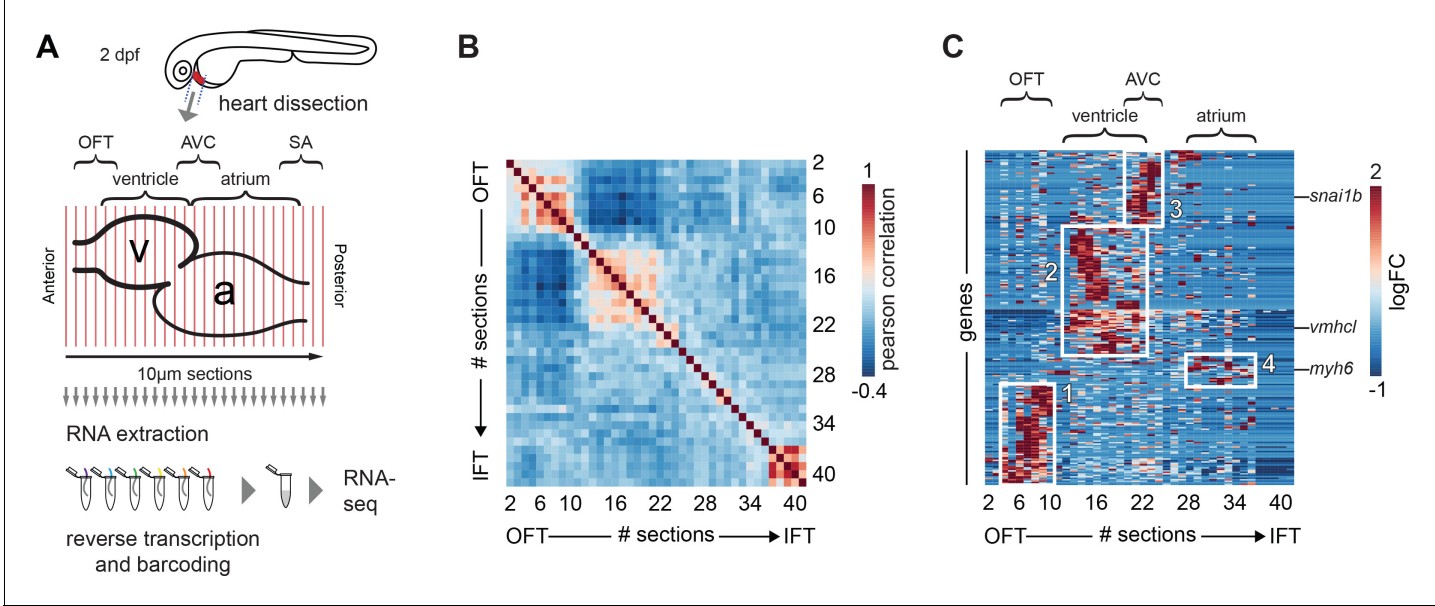

**Figure 1.** *Tomo*-seq on isolated embryonic hearts reveals distinct clusters of gene expression. (**A**) Hearts were isolated from 2-day-old embryos and 10 μm sections were made along the anterior-posterior axis, from outflow to inflow pole. Each section was collected in an individual tube followed by RNA isolation and cDNA transcription using section specific barcodes. After that, samples were pooled for linear amplification and sequence library preparation. (**B**) Pairwise correlation between individual sections across all genes detected at more than 20 reads in at least two sections. + 1 equals total positive correlation, 0 no correlation, −1 total negative correlation. Blocks of correlating sections can be observed. (**C**) Hierarchical cluster analysis of gene expression per section. Distinct gene expression clusters correspond to different regions of the heart shown in boxed areas: 1. arterial pole/OFT; 2. Ventricle; 3. AV canal; 4. Atrium. RNA sequencing reads per gene were normalized against the total read count per section as well as the total spike-in control RNA read counts. Genes expressed in min three sections z-value >1.25. See also *Figure 1—figure supplement 1* and *Figure 1—source data 1*.

DOI: https://doi.org/10.7554/eLife.31515.002

The following source data and figure supplement are available for figure 1:

**Source data 1.** Whole dataset table.
DOI: https://doi.org/10.7554/eLife.31515.004

**Figure supplement 1.** *Tomo*-seq statistics for the analyzed two dpf dataset.
DOI: https://doi.org/10.7554/eLife.31515.003

*source data 1*). To assess global transcriptome patterns in the dataset, we implemented Pearson's correlation analysis, a pairwise measurement of linear correlation per section across the total transcriptome. Several blocks of continuous sections with high positive correlation of overall gene expression were observed (*Figure 1B*). This indicates that specific molecular profiles based on explicit gene expression patterns subdivide the embryonic heart at this stage. In order to identify such gene clusters and elucidate underlying gene expression patterns, we performed a hierarchical clustering analysis. Robustly expressed genes (z-score >1.2, in three or more consecutive sections) were clustered according to their expression peak within the dataset. The cluster plot confirmed the presence of four large gene clusters with region-specific expression, which likely corresponded to the outflow tract, ventricle, AV canal and atrium based on the genes present within these clusters (*Figure 1C*).

To validate the presence of the above mentioned four cardiac sub-compartments and to identify the SA region where pacemaker cells reside in the tomo-seq data, we analyzed the expression of well-known cardiac genes. The total number of reads for a gene of interest was plotted against the consecutive section numbers and compared to its localized expression detected by whole mount in situ hybridization. Expression of *myosin light chain 7* (*myl7*) defining the boundaries of the myocardial tissue (*Figure 2A*), indicated that all sections with exception of the first ten, were derived from myocardial tissue. The first 10 sections most likely contained the non-myocardial arterial pole. *Natriuretic peptide a* (*nppa*) expression is restricted to the working myocardium of the ventricle and atrium and was absent from the non-working myocardium of the outflow tract, AV canal and SA

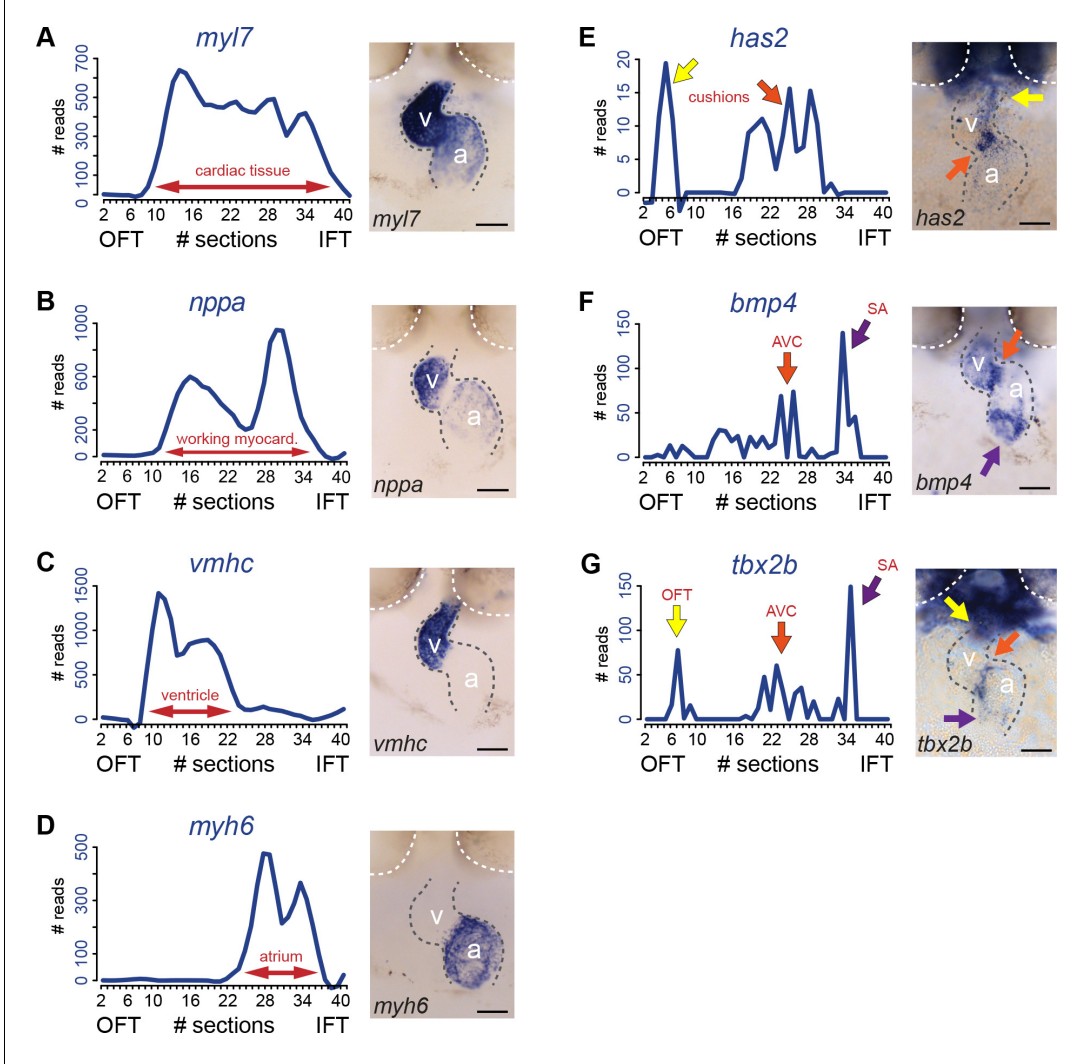

**Figure 2.** Transcriptome map of the embryonic heart with high spatial resolution. (A–E) *Tomo*-seq expression traces and corresponding in situ hybridization for (A) *myl7* (whole myocardium), (C) *vmhc* (ventricular myocardium), (D) *myh6* (atrial myocardium), (E) *has2* (endocardial cushions), (F) *bmp4* (AVC myocardium, orange arrow; IFT myocardium, purple arrow) and (G) *tbx2b* (OFT myocardium, yellow arrow, AVC myocardium, orange arrow; IFT myocardium, purple arrow). Smoothening (LOESS) was applied to graphs A-E, span α = 0.3. Anterior up. Gray dashed line outlines the heart. White dashed line outlines the eyes. A, atrium; V, ventricle; AVC: atrioventricular canal; IFT, inflow tract; OFT, outflow tract; SA, sinoatrial region. Scale bars represent 50 μm.

DOI: https://doi.org/10.7554/eLife.31515.005

region (*Figure 2B*). Together with the ventricle-specific expression of *ventricle myosin heavy chain* (*vmhc*) (*Figure 2C*) and atrial-specific *myosin heavy chain 6* (*myh6*) (*Figure 2D*) we concluded that the ventricle and the atrium were well separated in the *tomo*-seq data. To address whether genes that are expressed in the endocardial cushions, which give rise to the cardiac valves located in the outflow tract and in the AV canal, are present in the *tomo*-seq data, we analyzed the expression of *hyaluronan synthase 2* (*has2*) (*Figure 2E*). Indeed, *has2* was detected in the sections assigned to the outflow tract and the AV canal. The non-working myocardium of the AV canal can be distinguished by the expression of *bone morphogenetic protein 4* (*bmp4*) and *T-box protein 2b* (*tbx2b*). Both genes showed a peak expression around section 25, indicating the position of the AV myocardium (*Figure 2F and G*). Importantly, *bmp4* is also expressed in the SA region (*Figure 2F*). In the *tomo*-seq data, a clear expression peak was detected around section 34 (*Figure 2F*), indicating the position of the SA region in the *tomo*-seq data. We superimposed the expression data of the known marker genes to identify the position of the cardiac sub-compartments in the *tomo*-seq data

(*Figure 3*). We directly compared the superimposed expression data with the previously identified gene clusters and concluded that positions of cardiac sub-compartments correlated well with the identified gene clusters. The SA region was only identified by the marker gene expression and not by the gene cluster analysis, most likely due to its small size in combination with the threshold settings of the gene cluster analysis.

In conclusion, we have generated a genome-wide transcriptome dataset with high spatial resolution. Furthermore, the dataset reveals sub-compartments in the heart with distinct expression profiles.

## Identification of differentially expressed genes

To identify molecular pathways that may be regulating development of the cardiac sub-compartments, we analyzed the sub-compartment-specific transcriptomes in more detail. The boundaries of the sub-compartments were set based on the gene cluster analysis and the mRNA-seq reads for known marker genes (*Figure 3*). All genes were ranked according to the specificity of their expression level in the region of interest when compared with the rest of the dataset. With this approach, we identified a total of 1143 genes for which their expression was upregulated in one of the selected cardiac sub-compartments (ventricle, atrium, AV canal and sinoatrial region) (log2 FC >2 and p<0.05) (*Figure 4A–C* and *Figure 4—source data 1*). This list of differentially expressed genes contained several genes for which the sub-compartment expression has been well described, such as *myh6* (atrium), *vmhc* and *vmhcl* (ventricle), *spp1* (AV valves), and *isl1, bmp4 and shox2* (sinoatrial region). Besides these few known sub-compartment-specific marker genes, we identified many additional differentially expressed genes (99 genes for the atrium, 196 genes for the ventricle, 346 genes for the AV canal and 502 genes for the sinoatrial region). To validate the significance of the identified sub-compartment-specific gene-expression profiles in the *tomo*-seq data, we performed in situ hybridization for selected genes. Using this approach, the sub-compartment-specific gene expression profiles deduced from the *tomo*-seq data correlated well with the expression patterns obtained by in situ hybridizations (*Figure 4D–F* and *Figure 4—figure supplement 1A–B*).

We identified 99 genes with upregulated expression in the atrium and 196 genes with upregulated expression in the ventricle (*Figure 4A*). Gene ontology analysis revealed enrichment in *oxygen transport* and *tricarboxylic acid cycle* genes in the ventricle, suggesting that already in the 2-day-old ventricle metabolism is converting to aerobic oxidative

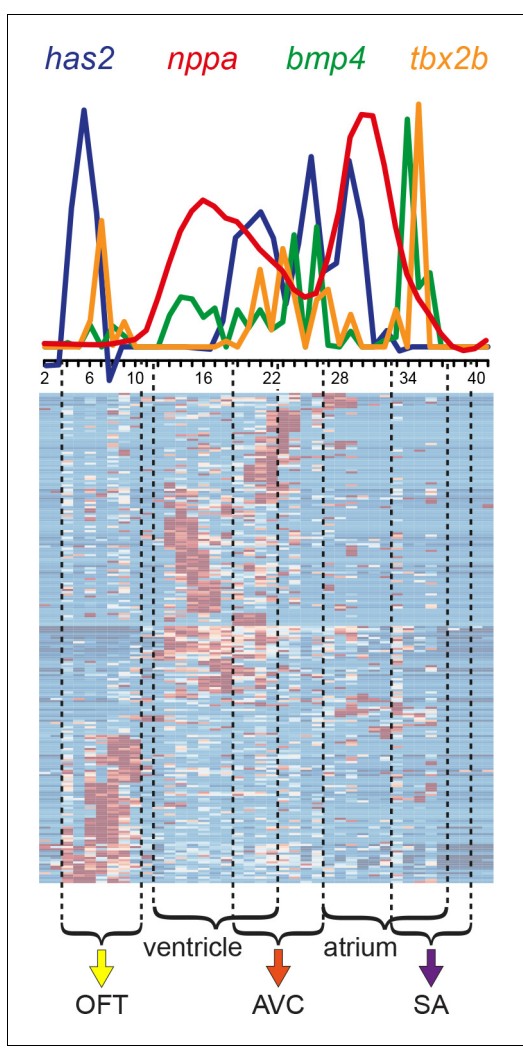

**Figure 3.** Differential gene expression reveals distinct sub-compartments within the heart. Superimposed expression profiles of *has2, nppa, bmp4* and *tbx2b*. Expression peaks of the individual genes overlap with the position of the clusters (dashed lines). *has2* and *tbx2b* in the OFT (yellow arrow). *nppa* in the ventricle. *has2, bmp4* and *tbx2b* in the AVC (orange arrow). *nppa* in the atrium. *bmp4* and *tbx2b* in the SA region (purple arrow). Smoothening (LOESS) was applied to *has2* and *nppa* graphs, span α = 0.3. OFT, outflow tract; AVC: atrioventricular canal; SA, sinoatrial region.
DOI: https://doi.org/10.7554/eLife.31515.006

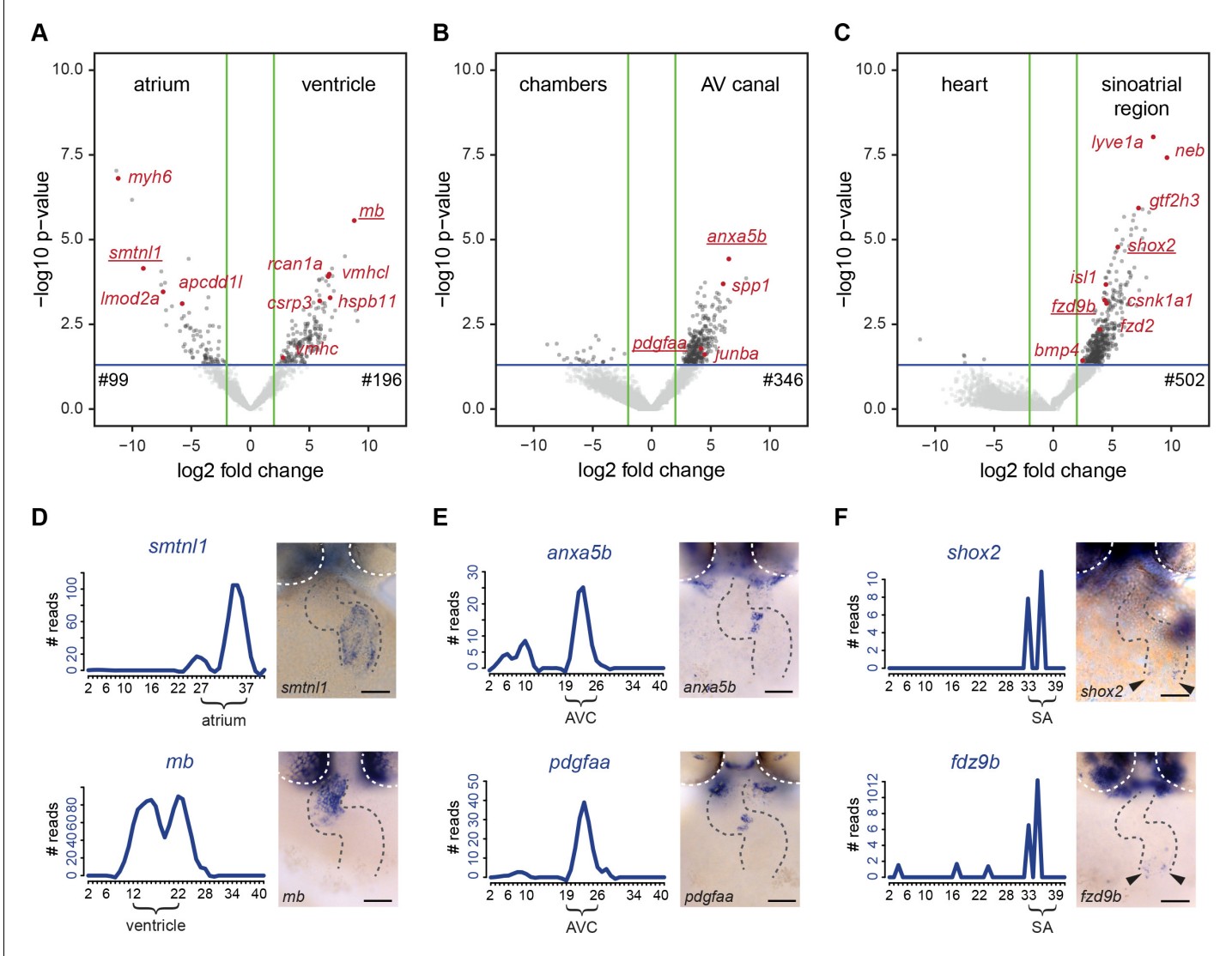

**Figure 4.** Differentially expressed genes in cardiac sub-compartments. (A–C) Volcano plots highlighting genes differentially expressed in the (A) atrium (n = 99) and ventricle (n = 196), (B) AV canal (n = 346) and (C) sinoatrial region (n = 502). Grey and red dots represent individual genes. Green lines indicate threshold of log2 fold change >2. Blue lines indicate threshold of p-value<0,05. (D–F) Expression traces and in situ hybridization analysis for representative example genes (gene names underlined in volcano plots) significantly upregulated in (D) atrium (*smtnl1*) and ventricle (*mb*), (E) AV canal (*anxa5b* and *pdgfaa*) and in the sinoatrial region (*shox2* and *fzd9b*). Smoothening (LOESS) was applied to graphs D and E, span α = 0.2. Gray dashed line outlines the heart. White dashed line outlines the eyes. Anterior up. Scale bars represent 50 µm. See also *Figure 4—figure supplement 1* and *Figure 4—source data 1*.

DOI: https://doi.org/10.7554/eLife.31515.007

The following source data and figure supplement are available for figure 4:

**Source data 1.** Genes significantly upregulated in cardiac sub-compartments.
DOI: https://doi.org/10.7554/eLife.31515.009
**Figure supplement 1.** Identification of region-specific gene expression patterns.
DOI: https://doi.org/10.7554/eLife.31515.008

phosphorylation to fulfill higher energy demands (*Figure 4—figure supplement 1C*). Furthermore, several genes related to cardiac myopathies were found amongst the ventricle specific genes (e.g. *csrp3*, *rcan1a* (*Figure 4—figure supplement 1A*), *tnnc2*, *myo1c* and *actn1*), indicating the value of the dataset for identifying potential disease-causing genes.

The AV canal region contains the AV myocardium and the endocardial cushions, forming the cardiac valves (*Beis et al., 2005*). We identified 346 differentially expressed genes in the AV canal, but gene-ontology analysis did not reveal any significant enrichment for specific biological processes. In situ hybridization confirmed the strong expression of *annexin A5* (*axna5b*) and *PDGF A* (*pdgfaa*) in the endocardial cushions. Both Annexin A5 and PDGF A have not been directly implicated in cardiac valve development, but PDGF A signaling plays numerous roles in organogenesis during embryo development by regulation of proliferation and directed cell migration (*Hoch and Soriano, 2003*).

The sinoatrial region contains the pacemaker and proepicardial cells. The majority (502) of the sub-compartment-specific upregulated genes was identified in this region. Gene ontology analysis revealed enrichment in *proepicardium development* genes based on differential expression of *tbx5a*, *acvr1l* and *bmp4* in the sinoatrial region (*Figure 4—figure supplement 1E*). In addition, several known regulators of pacemaker development (e.g. *isl1* and *shox2*) were identified in our analysis (*Figure 4C,F*). Interestingly, *hippo signaling* genes were enriched in the gene ontology analysis. Indeed, the *tomo*-seq dataset contained several components of active hippo signaling, *sav1*, *stk3* and *lats1/2* that were differentially expressed in the sinoatrial region. Moreover, we noticed that several components of the Wnt-signaling pathway were amongst the genes enriched in the sinoatrial region, such as *casein kinase 1, alpha 1* (*csnk1a1*), *frizzled homolog 2* (*fzd2*) and *frizzled homolog 9b* (*fzd9b*) (*Figure 4C,F* and *Figure 5—figure supplement 1*). Finally, the differential gene expression analysis identified *lymphatic vessel endothelial hyaluronic receptor 1a* (*lyve1a*) as being upregulated in the sinoatrial region. The *lyve1* gene is a marker for lymphatic endothelium. Further factors associated with the lymphatic system were expressed in the sinoatrial region (*prox1a*, *prox1b*, *nr2f1a*, *nr2f2*, *elmo1* and *foxj2*), suggesting that, as in the mammalian heart, cardiac lymphatics can develop from the sinus venosus (*Klotz et al., 2015*).

In conclusion, the *tomo*-seq dataset reveals cardiac sub-compartment-specific transcriptomes and has the potential to identify novel pathways that regulate the development or function of specific domains and structures within the heart.

## Wnt/β-catenin signaling activity in pacemaker cells

Since there is very little known about the regulation of pacemaker development, we focused our attention on the transcriptome of the sinoatrial region. As mentioned above, several genes that encode for Wnt signaling pathway components were enriched in the sinoatrial region. Wnt ligands bind to Frizzled (Fzd) receptors to activate intracellular β-catenin signaling, which plays multiple roles during cardiac development (reviewed in [*Gessert and Kühl, 2010*]). Depending on the developmental stage, Wnt/β-catenin signaling has either positive or negative effects on cardiomyocyte differentiation (*Tzahor, 2007*; *Gessert and Kühl, 2010*; *Ueno et al., 2007*). In addition, Wnt/β-catenin signaling in the atrioventricular (AV) canal induces the formation of endocardial cushions, giving rise to the AV valves, and the electrophysiological properties of the AV canal myocardium by slowing down electrical conductivity (*Hurlstone et al., 2003*; *Verhoeven et al., 2011*; *Gillers et al., 2015*). Even though Wnt/β-catenin signaling has been studied extensively in the heart, a role in pacemaker development has not been reported.

Expression of the Wnt receptor *fzd9b* was detected in sections 33–39 and whole mount in situ hybridization confirmed its expression in the sinoatrial region (*Figure 4F*). In addition, expression of the Wnt inhibitor *apcdd1l* was observed in the atrium, but its expression was absent from sections 33 to 39, defining the sinoatrial region (*Figure 4—figure supplement 1B*). From these results, we hypothesized that Wnt signaling is locally activated in the sinoatrial region. To test this further, we analyzed a Wnt-reporter line containing a transgene with 7 repeats of a TCF-binding site upstream of mCherry (*Moro et al., 2012*). To identify the Isl1[+] pacemaker cells, we made use of a transgenic *tg(Isl1:GFF;UAS:GFP)* line, further referred to as Isl1:GFP. Corroborating the hypothesis that Wnt/β-catenin signaling is activated in the sinoatrial junction, we observed cells that co-expressed both the *TCF:mCherry* reporter as well as the Isl1:GFP reporter in the sinoatrial region (*Figure 5A–D'*). In addition, TCF-mCherry-positive cells were also located in the AV canal, a cardiac sub-domain in which Wnt-signaling plays a known role (*Gillers et al., 2015*; *Hurlstone et al., 2003*; *Verhoeven et al., 2011*) (*Figure 5C*). Thus, Wnt-signaling is activated in the Isl1[+] cardiac pacemaker cells. In both zebrafish and mouse embryos lacking functional Isl1, a progressive failure of pacemaker function leads to severe bradycardia and arrhythmia (*de Pater et al., 2009*; *Tessadori et al., 2012*; *Liang et al., 2015*). Isl1-GFP[+] cells were still present in the sinoatrial region of *isl1* mutants

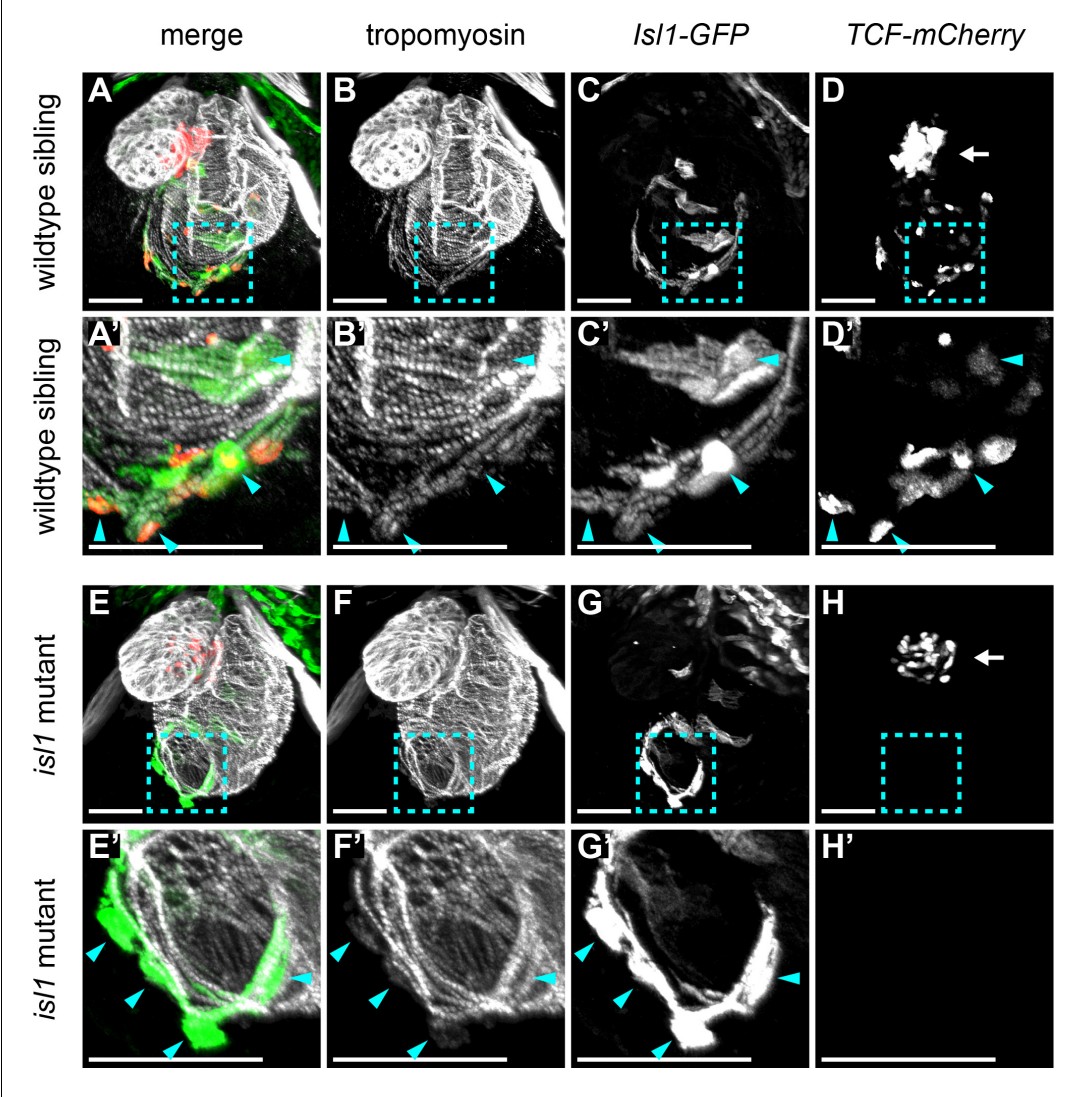

**Figure 5.** Wnt/β-catenin signaling in cardiac pacemaker cells. 3D reconstructions of confocal scans from whole mount embryos containing the *tg (7xTCFXla.Siam:nlsmCherry)* (reporting Wnt/β-catenin activity in red) and the *tg(Isl1:GFF;UAS:GFP)* (reporting isl1 expression and marking pacemaker cells in green) and stained for tropomyosin (myocardium in white). Anterior is up and posterior is down. Wild-type sibling (A-D') and *isl1^K88X* mutant (E-H') embryos at 3 dpf. The blue dashed line boxes in A-D and E-H indicate the sinoatrial region shown enlarged in A'-D' and E'-H', respectively. In wild-type hearts, Isl^+ pacemaker cells (blue arrowheads) co-expressed TCF-mCherry. In *isl1^-/-* mutants, no TCF-mCherry expression was observed in the Isl1^+ pacemaker cells (blue arrowheads). TCF-mCherry expression was also detected in the AV canal of wild type (D) and *isl1^-/-* mutants (H) (arrows). Scale bars represent 50 μm. See also *Figure 5—figure supplement 1*.

DOI: https://doi.org/10.7554/eLife.31515.010

The following figure supplement is available for figure 5:

**Figure supplement 1.** Expression profiles of Wnt-signaling components.

DOI: https://doi.org/10.7554/eLife.31515.011

(*Figure 5A–B' and E–F'*), suggesting that pacemaker cell specification still occurs in the absence of Isl1, but that these cells fail to differentiate into functional pacemaker cells. To determine whether Wnt-signaling requires Isl1, we also crossed the TCF-Cherry reporter into the *isl1* mutants. Interestingly, Wnt/β-catenin activity was lost specifically in the Isl1-GFP^+ cells of the *isl1* mutant embryos, while it was still detectable in the AV canal (*Figure 5C,C', G and G'*). These results reveal that in developing pacemaker cells active Wnt/β-catenin signaling depends on Isl1 activity.

# Myocardial loss of Wnt/β-catenin signaling affects autonomic control of heart rate

To assess the role of canonical Wnt signaling in pacemaker cells, we used a previously reported TetON system, in which Wnt/β-catenin signaling can be inhibited by the inducible and tissue-specific overexpression of its endogenous inhibitor Axin1 (*Knopf et al., 2010*). Overexpression of Axin1 stabilizes the β-catenin destruction complex, resulting in the inhibition of canonical Wnt signaling (*Nakamura et al., 1998*). Since cardiac specification and differentiation is regulated by Wnt/ β-catenin signaling at various stages during development (*Ueno et al., 2007*), it was important to block Wnt/β-catenin in a tissue-specific and temporally controlled manner. In the TetON system, Axin1 overexpression can be driven by the myocardial *myl7* gene promoter to restrict expression to the myocardium (*Knopf et al., 2010*). Since pacemaker cells also express the *myl7* gene, we reasoned that the TetON system could be used to inhibit Wnt signaling in the myocardium including the pacemaker cells without affecting embryo development in general. As a functional read-out for pacemaker function, we analyzed heart rates of control and Axin1 overexpressing embryos by high-speed video imaging and image analysis at 3 dpf (*Figure 6A*). Surprisingly and in contrast to the decreased heart rates observed in *isl1*[-/-] mutants (*Tessadori et al., 2012*; *de Pater et al., 2009*), inhibiting Wnt/β-catenin signaling by induction of Axin1 expression in the myocardium significantly increased cardiac contraction rates at 3 dpf (*Figure 6B,C*). Despite the 10–15% increase in heart rate, the beating pattern was still regular (see *Figure 6—video 1* and *Figure 6—video 2*, and *Figure 6—figure supplement 1*). To determine at which point in development active Wnt signaling is required to regulate heart rate, we induced Axin1 expression at different time points and measured the effect on heart rate at 3 dpf. Only Axin1 overexpression specifically between 36 hpf and 52 hpf, when the TCF:mCherry reported active Wnt signaling in the pacemaker cells, significantly increased heart rate (*Figure 6D*).

The autonomic nervous system controls heart rate by influencing the activity of the ion channels on the surface membrane through G-protein coupled β-adrenergic and cholinergic receptors (*Gordan et al., 2015*). These receptors are expressed in the Isl1[+] pacemaker cells (*Stoyek et al., 2016*) and can be stimulated by isoproterenol and carbachol respectively. We have previously observed similarly high heart rates in zebrafish embryos treated with isoproterenol (*Lodder et al., 2016*), suggesting that the observed increase in heart rate could be related to impaired autonomic control. To address this, we first induced Axin1 overexpression, followed by incubation of the embryos with either carbachol, a parasympathetic agonist, or isoproterenol, a sympathetic agonist (*Hsieh and Liao, 2002*; *Dlugos and Rabin, 2010*). While control embryos showed a significant reduction in heart rate after addition of carbachol, the reduction in heart rate was abolished in Axin1 overexpressing embryos (*Figure 6E–F*). Despite their already high heart rate, the Axin1-overexpressing embryos still responded to isoproterenol by increasing their heart rate further (*Figure 6E–F*). This resulted in extreme heart rates of up to 266 bpm in some of the embryos with ectopic Axin1 expression and incubated with isoproterenol. Together, these results indicate that Wnt/β-catenin signaling in the heart regulates the parasympathetic, but not the sympathetic control of heart rate. At the examined embryonic stage, the heart still lacks any direct innervation by the nervous system (*Figure 6—figure supplement 2*). However, *muscarinic cholinergic receptor 2a* (*chrm2a*) encoding for the muscarinic M2 receptor responsible for parasympathetic regulation of the heart rate is expressed in the sinoatrial region of the adult zebrafish heart (*Stoyek et al., 2016*) as well as the embryonic heart (*Steele et al., 2009*; *Thisse et al., 2004*). At 2 and 3 dpf, *chrm2a* was expressed in the sinoatrial region of the heart (*Figure 6—figure supplement 3A,B*). Overexpression of Axin1 resulted in a slight reduction in the expression of *chrm2a* (*Figure 6—figure supplement 3C*), indicating that Wnt/β-catenin can work at the level of the M2 receptor.

Together, our results reveal a new role for Wnt/β-catenin signaling in the autonomic control of heart rate by regulating the response of pacemaker cells to parasympathetic stimuli (*Figure 6G*).

## Discussion

Recently, transcriptome maps with spatial resolution were made of the murine heart (*DeLaughter et al., 2016*; *Li et al., 2016*). These maps were made by combining microdissection of embryonic hearts with single-cell mRNA-seq. Although these transcriptome maps contain single-cell expression data, the microdissection limited their spatial resolution. In addition, rare cell types may

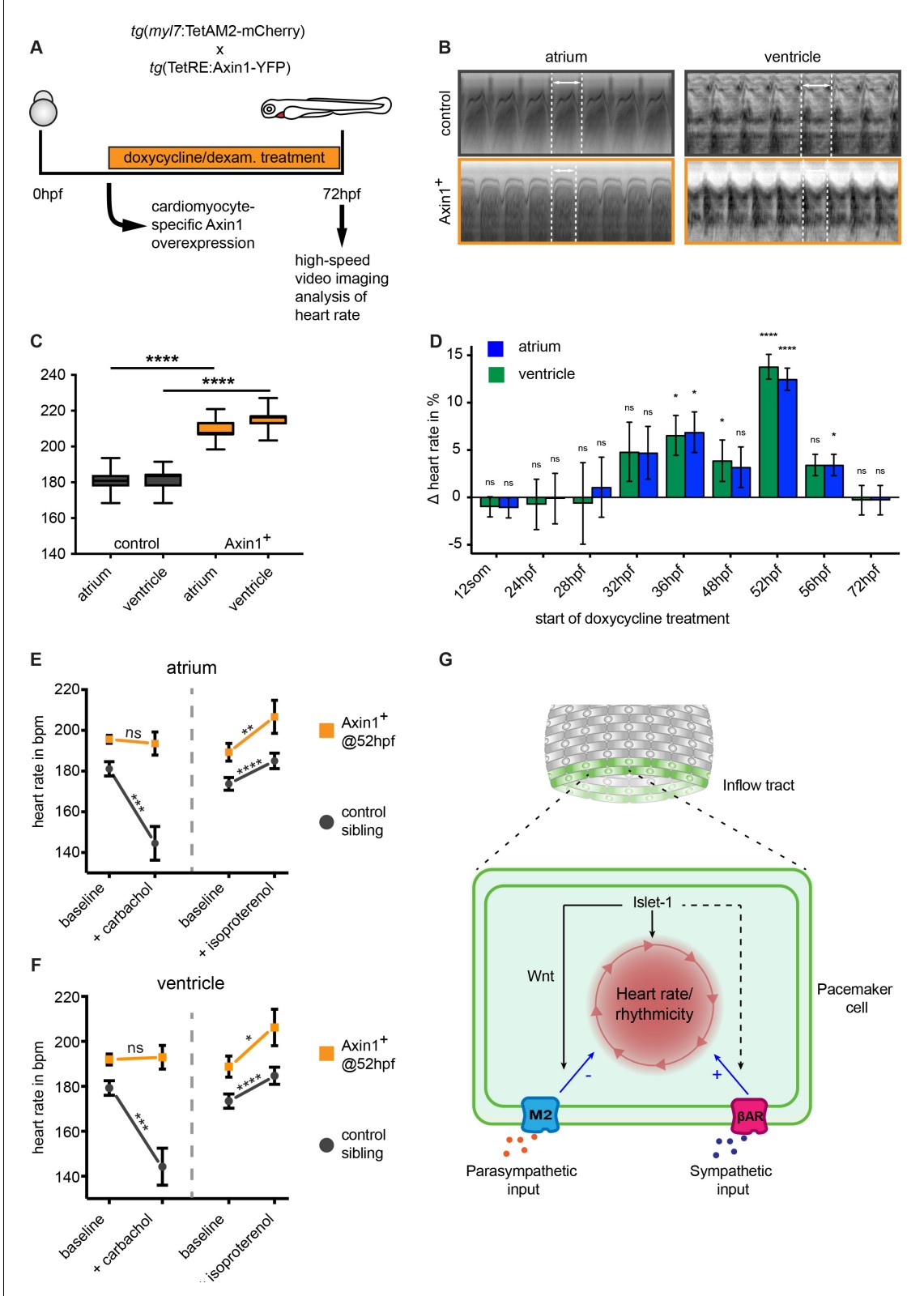

**Figure 6.** Wnt/β-catenin signaling regulates parasympathetic control of heart rate. (**A**) Experimental timeline. Embryos with the ubiquitous tg(TETRE: Mmu.Axin1-YFP)$^{tud1}$ and cardiomyocyte-specific tg(myl7:TETAM2-2A-mCherry)$^{ulm8}$ transgenes were incubated with doxycycline (25 μg/mL) plus dexamethasone (100 μM) to induce cardiomyocyte specific Axin1-YFP expression. Axin1-YFP was induced at different time points and the heart rate was measured at 3 dpf by high-speed video imaging and image analysis. (**B**) Atrial and corresponding ventricular kymographs from a representative

*Figure 6 continued on next page*

*Figure 6 continued*

mCherry⁻/YFP⁻ sibling control and an embryo with Axin1-YFP overexpression (Axin1⁺). Note the shorter period per full heartbeat (indicated by white dotted lines) in the Axin1 expressing embryo. Movies are available as *Figure 6—videos 1* and *2*, respectively. (C) Result of heart rate analysis (ventricle and atrium) on sibling and Axin1⁺ embryos, showing a significant increase in heart rate in Axin1⁺ embryos. (D) Relative changes in heart rate measured at 3 dpf after induction of Axin1 expression at various time points. Δheart rate was calculated by using the heart rates measured in the Axin1⁺ group and the control group using the following formula: (Axin1-YFP⁺ - mCherry⁻/YFP⁻) / (mCherry⁻/YFP⁻) x 100%. Induction of Axin1-YFP expression between 32 hpf and 56 hpf resulted in increased heart rates at 3 dpf. See also *Figure 6—videos 1* and *2. n(control/Axin1⁺)* 9/6 (12som); 20/15 (24hpf); 10/7 (28hpf); 18/16 (32hpf); 10/8 (36hpf); 30/21 (48hpf); 37/28 (52hpf); 27/23 (56hpf); 30/29 (72hpf). Column bar graph plotting mean with SEM. Atrial (green) and ventricular (blue) analysis (E–F) Heart rates at 3 dpf of control (mCherry⁻/YFP⁻) or Axin1-YFP⁺ embryos at baseline or after treatment with the parasympathetic agonist carbachol or the sympathetic agonist isoproterenol. Atrial (E) and ventricular (F) measurements. (left *n* = 27 (control), 26 (Axin⁺); right *n* = 20 (control), 14 (Axin⁺)) (G) Proposed model for Isl1/Wnt function in cardiac pacemaker cells. Isl1 is a central factor in pacemaker cell development. Isl1 expression is restricted to pacemaker cells in which it is required to establish the rhythmic cycle of membrane depolarizations and heart contractions and the activation of Wnt/β-catenin signaling. Wnt/β-catenin signaling is required to establish M2 cholinergic receptor signaling, which allows parasympathetic signals (e.g. acetylcholine) to adjust the rate of rhythmic membrane depolarizations and heart contractions. βAR, β-Adrenergic Receptor. Statistical significance (unpaired (D) and paired (E–F) t-test) *p<0.05; **p<0.01; ***p<0.001; ****p<0.0001. See also *Figure 6—source data 1*, *Figure 6—figure supplement 1*, *Figure 6—figure supplement 2*, *Figure 6—figure supplement 3*, *Figure 6—video 1* and *Figure 6—video 2*.

DOI: https://doi.org/10.7554/eLife.31515.012

The following video, source data, and figure supplements are available for figure 6:

**Source data 1.** Heart rate measurements and detailed statistics shown in *Figure 6C–F*.
DOI: https://doi.org/10.7554/eLife.31515.017
**Figure supplement 1.** Heart rate variability in Axin1⁺ embryonic hearts.
DOI: https://doi.org/10.7554/eLife.31515.013
**Figure supplement 1—source data 1.** Heart rate measurements and detailed statistics shown in *Figure 6—figure supplement 1*.
DOI: https://doi.org/10.7554/eLife.31515.014
**Figure supplement 2.** No neuronal innervation was detected in the 3 dpf heart.
DOI: https://doi.org/10.7554/eLife.31515.015
**Figure supplement 3.** M2 muscarinic receptor is expressed at the venous pole.
DOI: https://doi.org/10.7554/eLife.31515.016
**Figure 6—video 1.** Bright-field video recording of a mCherry⁻/YFP⁻ sibling heart at 3dpf.
DOI: https://doi.org/10.7554/eLife.31515.018
**Figure 6—video 2.** Bright field video recording of a Axin-YFP⁺ TetON heart at 3dpf.
DOI: https://doi.org/10.7554/eLife.31515.019

have been missed since not all cells within the heart were sequenced. In comparison, the *tomo*-seq transcriptome map was made by sectioning the entire heart in cryosections of 10 μm thickness. The high resolution and inclusion of all cardiac cells using the *tomo*-seq method likely explains why we were able to identify more differentially expressed genes even in a rare cell type such as the pacemaker cells. We established a comprehensive transcriptome map that includes the expression patterns of all genes expressed in the 2-day-old zebrafish heart. The 502 genes differentially expressed in the sinoatrial region encompass many uncharacterized ones, indicating the potential value of this dataset for revealing new pathways that regulate pacemaker development. A limitation of the *tomo*-seq transcriptome map is that the gene expression information is not cell-type specific, which can be resolved by additional in situ hybridizations or combining *tomo*-seq with single-cell mRNA sequencing.

Wnt/β-catenin signaling in the AV canal regulates the expression of *bmp4* and *tbx2* (*Verhoeven et al., 2011*). Although *bmp4* and *tbx2* are also expressed in the sinoatrial region, this expression does not depend on Wnt/β-catenin signaling ([*Verhoeven et al., 2011*] and data not shown). We found that Isl1 is required for the activation of Wnt/β-catenin signaling in the sinoatrial region (*Figure 5*). Since *isl1* is not expressed in the AV canal (*Hami et al., 2011*; *Tessadori et al., 2012*), and since we observed active Wnt/β-catenin signaling in the AV canal of *isl1⁻/⁻* mutants (*Figure 5G*), we conclude that Wnt/β-catenin signaling plays different roles in the AV canal and sinoatrial region.

We found that Wnt/β-catenin signaling activity is required between 36 hpf and 52 hpf to reduce the heart rate (*Figure 6D*). This was unexpected since *isl1⁻/⁻* mutant hearts have a severely reduced heart rate (*de Pater et al., 2009*; *Tessadori et al., 2012*). To explain these seemingly contradicting

observations we propose a model in which Isl1 has a very central role in establishing pacemaker cell differentiation, while Wnt has a more specific downstream role during pacemaker function (*Figure 6G*). We hypothesize that Isl1 regulates the expression of factors that are essential to drive the rhythmic membrane depolarizations, which is specific for pacemaker cells and absent from atrial cardiomyocytes. Without Isl1 the heart beat is maintained at low frequency since embryonic cardiomyocytes are able to spontaneously depolarize their membrane and contract, explaining the low beating frequency observed in *isl1*$^{-/-}$ mutants (*de Pater et al., 2009*; *Tessadori et al., 2012*). When the cardiomyocytes mature, and lose their ability to spontaneously contract, the heartbeat of *isl1*$^{-/-}$ mutants stops since the rhythmic membrane depolarizations were not established. In addition, Isl1 induces Wnt/β-catenin signaling in the pacemaker cells to establish parasympathetic control of the rhythmic membrane depolarizations. Since parasympathetic input through acetylcholine and muscarinic cholinergic receptors (mAChRs) decreases the heart rate, an increase in heart rate was observed after inhibiting Wnt/β-catenin signaling. Furthermore, we observed that carbachol, an acetylcholine agonist, was unable to reduce heart rate in embryos in which Wnt/β-catenin signaling was inhibited, indicating regulation at the level of the receptor or of downstream signaling. The M2 mAChR is expressed in the zebrafish heart from 30 hpf onwards and by 3dpf the autonomic response is mature (*Hsieh and Liao, 2002*; *Dlugos and Rabin, 2010*; *Shin et al., 2010*). Expression of the M2 mAChR was reduced after inhibiting Wnt/β-catenin signaling (*Figure 6—figure supplement 3A*), suggesting that Wnt/β-catenin signaling effects M2 receptor levels.

In summary, the spatially resolved transcriptome map of the embryonic heart provides an unprecedented opportunity to uncover novel mechanisms of cardiac development. The here identified Wnt-dependent mechanism to establish autonomous control of heart rate could be beneficial to ongoing efforts in creating biological pacemakers, which currently lack parasympathetic control (*Protze et al., 2017*).

# Materials and methods

**Key resources table**

| Reagent type (species) or resource | Designation | Source or reference | Identifiers | Additional information |
|---|---|---|---|---|
| Gene (*Danio rerio*) | *mb* | NA | ZDB-GENE-040426–1430 | |
| Gene (*D. rerio*) | *smtnl1* | NA | ZDB-GENE-050306–23 | |
| Gene (*D. rerio*) | *anxa5b* | NA | ZDB-GENE-030131–9076 | |
| Gene (*D. rerio*) | *pdgfaa* | NA | ZDB-GENE-030918–2 | |
| Gene (*D. rerio*) | *fzd9b* | NA | ZDB-GENE-000906–4 | |
| Gene (*D. rerio*) | shox2 | NA | ZDB-GENE-040426–1457 | |
| Strain, strain background (*D. rerio*) | Tupfel long fin (TL) | ZIRC,Eugene, OR | ZDB-GENO-990623–2 | |
| Genetic reagent (*D. rerio*) | Tg(UAS:GFP) | Kawakami lab Asakawa 2008 | ZDB-FISH-150901–15231 | |
| Genetic reagent (*D. rerio*) | Tg(Isl1:GFF;UAS:GFP) hu10018tg | this paper | | see Material and methods section |
| Genetic reagent (*D. rerio*) | Tg(7xTCFXla.Siam: nlsmCherry) | this paper | | The tg(7xTCF-nlsmCherry) wnt-signaling reporter line was established in a TL wildtype background using the pDEST-7xTCF-nlsmCHERRY-polyA construct (*Moro et al., 2012*) received from the Smith lab (IMB, Brisbane, Australia). |
| Genetic reagent (*D. rerio*) | isl1 mutant line. isl1sa29 | ZIRC, Eugene, OR | ZDB-GENO-990623–2 | |

*Continued on next page*

*Continued*

| Reagent type (species) or resource | Designation | Source or reference | Identifiers | Additional information |
|---|---|---|---|---|
| Genetic reagent (*D. rerio*) | Tg(myl7:EGFP): twu26tg | ZIRC, Eugene, OR | ZDB-FISH-150901–23018 | |
| Genetic reagent (*D. rerio*) | Tg(TETRE:Mmu.Axin1-YFP) tud1: tud1Tg | ZIRC, Eugene, OR, Gilbert Weidinger, Knopf 2012 | ZDB-FISH-150901–26117 | |
| Genetic reagent (*D. rerio*) | Tg(myl7:TETAM2-2A-mCherry)ulm8: ulm8Tg | ZIRC, Eugene, OR, Gilbert Weidinger, Knopf 2013 | ZDB-ALT-151214–5 | |
| Antibody | Anti-GFP (Chicken Antibodies, IgY Fraction) | Aves Labs, Tigard, OR | Cat# GFP-1010; RRID: AB_2307313 | 1:500 |
| Antibody | Living Colors DsRed Polyclonal Antibody | Clontech, Mountain View, CA | Cat# 632496; RRID: AB_10013483 | 1:200 |
| Antibody | Monoclonal Anti-Acetylated Tubulin antibody | Sigma-Aldrich | Cat# T7451; RRID: AB_609894 | |
| Antibody | Monoclonal Anti-Tropomyosin (Sarcomeric) antibody | Sigma-Aldrich | Cat# T9283; RRID: AB_261817 | 1:200 |
| Recombinant DNA reagent | BAC clone CH211-219F7 | BACPAC Resources Center (BPRC), Oakland, CA | ZDB-BAC-050218–781 | |
| Recombinant DNA reagent | pDEST-7xTCF-nlsmCHERRY-polyA | *Moro et al. (2012)* provided by Kelly Smith | ZDB-TGCONSTR CT-110113–2 | |
| Commercial assay or kit | TruSeq small RNA sample prep Kit | Illumina, San Diego, CA | RS-200–0012 or −0024, −0036, −0048 | |
| Commercial assay or kit | MessageAmpII Kit | Ambion, Foster City, CA | AM1751 | |
| Chemical compound, drug | Doxycycline hyclate | Sigma-Aldrich | D9891; CAS:24390-14-5 | |
| Chemical compound, drug | Dexamethasone | Sigma-Aldrich | D1756 CAS:50-02-2 | |
| Chemical compound, drug | Isoprenaline hydrochloride | Sigma-Aldrich | I5627; CAS:51-30-9 | |
| Chemical compound, drug | Carbamoylcholine chloride | Sigma-Aldrich | C4382; CAS:51-83-2 | |
| Software, algorithm | RStudio | | https://www.rstudio.com/ RRID:SCR_000432 | |
| Software, algorithm | Imaris data visualization software | Bitplane, Zürich, Switzerland | http://www.bitplane.com/imaris/imaris RRID:SCR_007370 | |
| Software, algorithm | HoKaWo Image acquisition module | Hamamatsu, Shizuoka, Japan | https://www.hamamatsu.com/eu/en/product/alpha/I/5006/U9304/index.html | |
| Other | Resource website with the complete 2dpf heart dataset | This paper | http://zebrafish.genomes.nl/tomoseq/ | |

## Contact for reagent and resource sharing

Further information and requests for reagents may be directed to, and will be fulfilled by the Lead Contact, Dr. Jeroen Bakkers, j.bakkers@hubrecht.eu.

## Experimental model and subject details

### Zebrafish

Fish used in this study were kept in standard conditions as previously described (*Westerfield, 1995*). Tupfel long fin (TL) wildtype, *isl1K88X (isl1^sa29/sa29*) (*de Pater et al., 2009*) mutant and *tg(myl7: EGFP)^twu26* (*Huang et al., 2003*) transgenic zebrafish lines were available from ZIRC, Eugene, OR. The TetON lines, ubiquitously expressed *tg(TETRE:Mmu.Axin1-YFP)^tud1* and cardiomyocyte-specific *tg(myl7:TETAM2-2A-mCherry)^ulm8* were received from the Weidinger lab (Ulm University, Germany) and crossed to obtain double transgenic embryos for wnt knockdown experiments. The Islet-1 reporter line *tg(Isl1:GFF;UAS:GFP)^hu10018tg* is an independent allele of a previously published line (*Tessadori et al., 2012*). Details on the generation of the transgenic line in *tg(UAS:GFP)* embryos (*Asakawa et al., 2008*) are given below. The *tg(7xTCF-nlsmCherry)* wnt-signaling reporter line was established in a TL wild-type background using the pDEST-7xTCF-nlsmCHERRY-polyA construct (*Moro et al., 2012*) received from the Smith lab (IMB, Brisbane, Australia). All studies involving vertebrate animals were performed with institutional approval in compliance with institutional ethical guidelines.

## Method details

### Tomo-sequencing

TL wildtype hearts were manually dissected from live 2-day-old zebrafish embryos. The heart tube was quickly transferred to Jung tissue freezing medium (Leica), carefully straightened and frozen on dry ice. The heart samples were cryo-sectioned in anterior to posterior direction with a thickness of 10 µm per section. Total RNA was isolated using Trizol (Ambion, Foster City, CA). An ERCC RNA spike-in mix (Ambion, Foster City, CA) was added for normalization. The sections were processed individually, barcoded and processed for RNA sequencing as previously described in detail (*Kruse et al., 2016*; *Junker et al., 2014*). Illumina sequencing libraries were generated and sequenced using the NextSeq platform.

### Data analysis

Independent tomo-seq datasets were established for three wild-type hearts. Based on statistical parameters (total read count, mappability of raw reads, tissue orientation and spatial separation of the two main chambers) and expression peaks for known cardiac genes, the most robust heart sample (heart #1) was chosen for further detailed analysis. All three datasets can be accessed via the tomo-seq website (http://zebrafish.genomes.nl/tomoseq/Burkhard2017/).

The raw data is accessible under GSE104057. (Password for reviewer access: gbsniksgjfunvcb).

Bioinformatical analyses were performed with R software (R Core Team, 2013) using custom-written code (see source code file). The individual gene read counts were first normalized against the total counts per section before re-normalization to the median of the total reads per section. Thus, ensuring rough equivalence between read count numbers and original number of mapped raw reads. In addition, expression reads were normalized to the total spike-in read count to limit technical bias. The number of sequencing reads mapped for individual genes were plotted per section. LOESS (locally weighted scatterplot smoothening) with a smoothening parameter (span) of $\alpha = 0.2$ (*Figure 4*, *Figure 4—figure supplement 1*) or $\alpha = 0.3$ (*Figure 2*, *Figure 3*, *Figure 5—figure supplement 1*) was applied.

For Pearson's correlation analysis, all genes expressed at >4 reads in >1 section were selected prior to total-read-normalization. Based on the log2-fold-change (zlfc) of the $Z$ score of all genes, correlation was calculated across the transcriptome for each pairwise combination of sections. Hierarchical clustering analysis on the entire dataset (after $Z$ score transformation) was performed on all genes with a peak in >3 consecutive sections ($Z$ score >1.2). Comparative analysis of regional gene expression profiles was done using the EdgeR package (*Robinson et al., 2010*). Regions of interest were determined by expression of known cardiac marker genes. Genes significantly upregulated in ROI (log2 fold change >2; p-value<0.05) were retrieved from the dataset and used for GO term enrichment analysis using the DAVID functional annotation tool (*Huang et al., 2009b*; *Huang et al., 2009a*).

## In-situ hybridization

For in-situ hybridization (ISH) TL wild-type embryos were fixed in 4% PFA-PBS overnight and dehydrated in methanol. ISH on 2-3 dpf embryos was carried out as previously described (*Westerfield, 1995*).

## Generation of the Islet-1 transgenic line

The *tg(Isl1:GFF$^{hu10018tg}$)* line was generated essentially as described previously (*Tessadori et al., 2012*; *Bussmann and Schulte-Merker, 2011*). An iTOL2_amp cassette for pTarBAC was inserted in the vector sequence of bacterial artificial chromosome (BAC) CH211-219F7, containing the full *isl1* locus. An expression cassette containing the GalFF gene (*Asakawa et al., 2008*) and kanamycin resistance gene was integrated at the ATG site of the 1 st exon of the isl1 gene. Primers used were (in lower case sequence homologous to BAC): isl1_Gal4FF_F 5'-gggccttctgtccggttttaaaagtggacctaa-cac cgccttactttcttaccATGAAGCTACTGT-CTTCTATCGAAC-3' and isl1_KanR_R 5'-aaataaacaa-taaagcttaacttacttttcggtggatcccccatgtctccTCAGAAGAACTCGTC

AAGAAGGCG-3'. Red/ET recombination was done following the manufacturer's protocol (Gene Bridges) with minor modifications. BAC-DNA isolation was carried out using a Midiprep kit (Life Technologies). 300 ng/µl of BAC-DNA was injected in *tg(UAS:GFP$^{nkuasgfp1aTg}$)* embryos (*Asakawa et al., 2008*) in combination with 25 ng Tol2 mRNA. To establish the stable transgenic line, healthy embryos displaying robust isl1-specific fluorescence were selected and grown to adulthood.

## Immunohistochemistry and image processing

3 dpf TL wild-type embryos were fixed in 2% PFA-PBS overnight. Immunohistochemistry was carried out as previously described (*de Pater et al., 2009*). The primary antibodies used were chicken αGFP (Aves Labs, Tigard, OR, RRID: AB_2307313, 1:500), rabbit αDsRed (Clontech, Mountain View, CA, AB_10013483, 1:200), mouse Anti-Acetylated Tubulin (Sigma-Aldrich (St. Louis, MO), AB_609894) and mouse αTropomyosin (Sigma, AB_261817, 1:200). All immunohistochemistry images are 3D reconstructions of confocal scans. Image processing was done using the Imaris data visualization software (Bitplane, Zürich, Switzerland). For image clarity, non-cardiac expression of the TCF-mCherry reporter line was removed (*Figure 5*).

## Drug treatments in zebrafish embryos

Doxycycline (Sigma, D9891) was dissolved in 50% EtOH, kept as a 50 mg/ml stock solution and stored at −20°C in the dark. Dexamethasone (Sigma, D1756) was dissolved in 100% EtOH, kept as a 10 mg/ml stock solution and stored at −20°C in the dark. For induction of the TetON system, we used a final concentration of 25 µg/ml of doxycycline and 100 µM dexamethasone diluted in E3 medium. Embryos were treated in 6 cm dishes with a maximum of 30 embryos and kept at 28°C in the dark during the drug incubation. Efficient induction was confirmed by cardiomyocyte-specific YFP expression immediately prior to the heart rate analysis.

Carbamoylcholine chloride (*Carbachol*, Sigma, C4382) was dissolved in mQ water at a stock concentration of 50 mM. Isoprenaline hydrochloride (*Isoproterenol*, Sigma, I5627) was dissolved in mQ water at a stock concentration of 50 mM. Stock solutions were kept at −20°C for up to 1 week. A final concentration of 500 µM Carbachol or 100 µM Isoproterenol was used in 12 ml E3 medium.

## High-speed imaging and analysis

Embryos were mounted in 0.25% agarose (Life Technologies BV) prepared in E3 medium with 16 mg/ml tricaine. Recording of embryonic hearts were performed with a high-speed CCD camera and HoKaWo Image acquisition module (Hamamatsu Photonics) at 150fps on a brightfield microscope (Leica). The temperature was kept at 28.5°C throughout the heart rate measurements using a controlled temperature chamber. Each heart was imaged for 10 s, capturing 30–40 consecutive cardiac cycles. For the carbachol/isoproterenol experiments, embryos were imaged before drug addition and after 30 min drug incubation to calculate relative change in heart rate. Image analysis was carried out with ImageJ (http://rsbweb.nih.gov/ij/). Statistical analysis was carried out using GraphPad Prism7 (GraphPad software).

## Quantification and statistical analysis

### Bioinformatical analyses

Sequencing reads obtained from the paired-end Illumina sequencing were mapped to the zebrafish genome (Ensembl genome assembly Zv9) as described previously (*Junker et al., 2014*). All sequencing data analysis including normalization, single gene expression analysis, identification of expression patterns and clustering was performed in R (http://www.R-project.org/) using custom-written code (Source code) as well as publicly available packages (EdgeR).

The heart rate analyses were performed in ImageJ by manual measurement of each cardiac cycle. Box-whisker plots, outlier removal and statistical analysis was done in GraphPad Prism7 (GraphPad software). Further statistical details of experiments can be found in the figure legends.

## Data and software availability

### Software

All software used in this study was obtained from publicly or commercially available resources.

### Data resources

The processed tomo-seq dataset used is publicly available via the website: http://zebrafish.genomes.nl/tomoseq/Burkhard2017/

The raw data can be retrieved from the GEO database (GSE104057).

## Acknowledgements

We thank Utrecht Sequencing Facility for providing the sequencing service and data. Utrecht Sequencing Facility is subsidized by the University Medical Center Utrecht, Hubrecht Institute and Utrecht University. We thank Fabian Kruse for bioinformatics support, Laurence Garric for discussions during the project, Federico Tessadori for help with the BAC transgenesis, Kelly Smith for help with the *tg(7xTCF-nlsmCherry)* analysis, the Hubrecht Imaging Facility for microscopy support and Life Science Editors for editing support. We acknowledge support from the Netherlands Cardiovascular Research Initiative, Dutch Heart Foundation grant CVON2014-18 CONCOR-GENES, CVON Predict, ZonMW grant 40-00812-98-12086, and the ERA-NET Cofund action N° 643578 under the European Union's Horizon 2020 research and innovation programme and national funding organisations Canadian Institutes for Health Research (CIHR), the Netherlands Organization for Health Research and Development (ZonMw), Belgium (Flanders) Research Foundation Flanders (FWO), and French National Research Agency (ANR) .

## Additional information

### Funding

| Funder | Grant reference number | Author |
|---|---|---|
| Nederlandse Organisatie voor Wetenschappelijk Onderzoek | 022.001.003 | Silja Barbara Burkhard |
| CVON - Netherlands Heart Foundation | CVON-CONCORgenes | Jeroen Bakkers |
| ZonMw | 91212086 | Jeroen Bakkers |
| ZonMw | 9003037607 | Jeroen Bakkers |

The funders had no role in study design, data collection and interpretation, or the decision to submit the work for publication.

### Author contributions

Silja Barbara Burkhard, Formal analysis, Validation, Investigation, Visualization, Writing—original draft; Jeroen Bakkers, Conceptualization, Supervision, Funding acquisition, Writing—review and editing

## Author ORCIDs

Silja Barbara Burkhard (ID) http://orcid.org/0000-0002-5761-4674
Jeroen Bakkers (ID) http://orcid.org/0000-0002-9418-0422

## Ethics

Animal experimentation: All studies involving vertebrate animals were performed with institutional approval in compliance with institutional ethical guidelines. (KNAW DEC 14-01)

## Decision letter and Author response

Decision letter https://doi.org/10.7554/eLife.31515.025
Author response https://doi.org/10.7554/eLife.31515.026

# Additional files

## Supplementary files

• Source Code 1. R source code R script code used to process, analyze and visualize the tomo-seq datasets.
DOI: https://doi.org/10.7554/eLife.31515.020

• Transparent reporting form
DOI: https://doi.org/10.7554/eLife.31515.021

## Major datasets

The following dataset was generated:

| Author(s) | Year | Dataset title | Dataset URL | Database, license, and accessibility information |
|---|---|---|---|---|
| Burkhard SB, Bakkers J | 2017 | Spatially resolved RNA-sequencing of the embryonic zebrafish heart | https://www.ncbi.nlm.nih.gov/geo/query/acc.cgi?acc=GSE104057 | Publicly available at the NCBI Gene Expression Omnibus (accession no: GSE104057) |

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
