## [Decision Letter]

Thank you for submitting your article "Spatially resolved RNA-seq of the embryonic heart identifies a role for Wnt signaling in autonomic control of heart rate" for consideration by *eLife*. Your article has been reviewed by two peer reviewers, and the evaluation has been overseen by a Reviewing Editor and Marianne Bronner as the Senior Editor. The reviewers have opted to remain anonymous.

The reviewers have discussed the reviews with one another and the Reviewing Editor has drafted this decision to help you prepare a revised submission.

Summary:

In this manuscript, Burkhard and Bakkers illustrate the use of tomo-seq to generate a high-resolution spatial transcriptome map of the developing zebrafish heart. The authors identified a large number of genes that are differentially expressed within sub-compartments of the heart, thereby creating a valuable resource for future analyses of cardiac regionalization. To demonstrate the utility of this resource, the authors looked closely at genes expressed in the sinoatrial region and noted the expression of Wnt receptor *fzd9b* in this area. They then demonstrated that Wnt signaling is active in the sinoatrial/pacemaker region and is controlled by Islet-1 activity. Finally, functional analysis elucidated that the Wnt signaling pathway controls embryonic heart rate through modulation of pacemaker cell response to parasympathetic stimuli.

This work holds interest for investigators studying cardiac development, as little is known about genes regulating early specification of the sinoatrial node. The experiments are performed in a technically sound and elegant manner, the data are of high quality, and sufficient detail is included. Thus, this manuscript contributes significantly to our understanding of how proper cardiac physiology is established during embryogenesis, and it would appeal to the readership of *eLife*. However, a few key issues remain that need to be addressed in order to strengthen the paper's conclusions.

Essential revisions:

1) An exciting finding of this study is that Wnt signaling regulates the parasympathetic response in the sinoatrial region of the developing heart. Has the heart been innervated at this early developmental stage? If not, what might be the source of the parasympathetic signal? Are there hints from the tomo-seq dataset?

2) The model for Isl1 function is somewhat confusing as it separates Isl1's role into two discrete activities: regulating the expression of genes that maintain rhythmic membrane depolarizations and regulating Wnt signaling/parasympathetic controls. Might it be more clear to discuss the function of Isl1 in regulating Wnt signaling as a component of its larger role in establishing pacemaker cell differentiation?

3) The techniques employed to measure heart rate (Figure 6) and heart rate variability (Figure 4—figure supplement 1) are less rigorous than might be expected. First, use of an atrial kymograph as opposed to a ventricular kymograph seems confusing. Additionally, manual selection of a region of interest in the kymograph can alter heart rate calculations, and the duration of the video image acquired (2.8 seconds) is rather short. Evaluation of heart rate from both chambers would be helpful, as would measurement of fractional shortening. Also, more accurate representations of heart rate variability such as standard deviation of the mean R-R interval would improve the accuracy of the data presented.

4) In Figure 6—figure supplement 3, *chrm2a* expression following Wnt inhibition appears downregulated although the authors state that there is no difference when compared to wild-type. Representation of the total number of embryos examined and the number depicting normal versus slightly decreased expression in the sinoatrial node region would be helpful in this context. Alternatively, a more representative image may be appropriate if expression levels are unchanged.

---

## [Author Response]

Essential revisions:1) An exciting finding of this study is that Wnt signaling regulates the parasympathetic response in the sinoatrial region of the developing heart. Has the heart been innervated at this early developmental stage? If not, what might be the source of the parasympathetic signal? Are there hints from the tomo-seq dataset?

To address this question we performed anti-acetylated tubulin antibody labelling combined with confocal microscopy to highlight potential innervating axons at 3 dpf (time point at which we did the functional analysis). From our analysis, no neuronal innervation of the cardiac tissue was detected at this stage, although superficial sensory neurons are clearly present. Since we believe that this information is important for the interpretation of the functional analysis we have now included a new supplementary figure presenting these data (Figure 6—figure supplement 3). In addition, we added a short statement in the text to explain this:

“At the examined embryonic stage the heart lacks any direct innervation by the nervous system (Figure 6—figure supplement 3).”

We do not know what the source is of the parasympathetic stimuli (e.g. acetylcholine) to which the pacemaker cells respond. Acetylcholine is secreted by neurons and is also present in the serum. We speculate that in the embryo, circulating acetylcholine can activate the M2 receptor in pacemaker cells.

2) The model for Isl1 function is somewhat confusing as it separates Isl1's role into two discrete activities: regulating the expression of genes that maintain rhythmic membrane depolarizations and regulating Wnt signaling/parasympathetic controls. Might it be more clear to discuss the function of Isl1 in regulating Wnt signaling as a component of its larger role in establishing pacemaker cell differentiation?

We agree with this comment that our suggestion of ‘two discrete activities’ is confusing. From our results, we conclude that Isl1 is required for both the establishment and regulation (through Wnt) of the rhythmic membrane depolarizations in the pacemaker cells. We agree that these are not two discrete activities of Isl1 but are both part of a larger role of isl1. To better show that Wnt signalling is a component of the larger role of Isl1 in pacemaker differentiation, we have adjusted the model in Figure 6. In addition we adjusted the text in the Discussion, which reads now as follows:

“To explain these seemingly contradicting observations we propose a model in which Isl1 has a very central role in establishing pacemaker cell differentiation, while Wnt has a more specific downstream role during pacemaker function (Figure 6). […] Since parasympathetic input through acetylcholine and muscarinic cholinergic receptors (mAChRs) decreases the heart rate, an increase in heart rate was observed after inhibiting Wnt/β-catenin signaling.”

3) The techniques employed to measure heart rate (Figure 6) and heart rate variability (Figure 4—figure supplement 1) are less rigorous than might be expected. First, use of an atrial kymograph as opposed to a ventricular kymograph seems confusing. Additionally, manual selection of a region of interest in the kymograph can alter heart rate calculations, and the duration of the video image acquired (2.8 seconds) is rather short. Evaluation of heart rate from both chambers would be helpful, as would measurement of fractional shortening. Also, more accurate representations of heart rate variability such as standard deviation of the mean R-R interval would improve the accuracy of the data presented.

We are sorry for the misunderstanding related to the method that we used to establish heart rates. Although in Figure 6 kymographs are shown representing only 2.8 sec, the actual kymographs used for the analysis had a duration of 10 seconds. The kymograph in Figure 6 was cropped to 2.8 seconds for better illustration. This was unclear from the description in the Materials and methods section and has been corrected.

For the analysis, we selected a region close to the pacemaker region to allow us to focus on pacemaker function while limiting a possible effect of slower conduction velocity or impaired contraction on our measurements. We have now reanalysed all the videos and repeated the heart rate analysis using ventricular kymographs as suggested and added the results to Figure 6. The new measurements taken from the ventricle are very similar to the original atrial measurements and therefore do not alter any of our conclusions. The measurements from both chambers are now included in the new Figure 6. We have not quantified fractional shortening due to the large variation observed in control embryos.

As suggested we have now also included the standard deviation of the mean R-R interval in Figure 6—figure supplement 1.

4) In Figure 6—figure supplement 3, chrm2a expression following Wnt inhibition appears downregulated although the authors state that there is no difference when compared to wild-type. Representation of the total number of embryos examined and the number depicting normal versus slightly decreased expression in the sinoatrial node region would be helpful in this context. Alternatively, a more representative image may be appropriate if expression levels are unchanged.

We appreciate this comment. We carefully re-examined *chrm2a* expression and agree that there was a mild but consistent reduction in the expression level of *chrm2a* in embryos with Axin1 overexpressing. We have quantified the results and included the total number of embryos examined and the number of embryos with the reduced expression.

Based on these results we have adjusted the description:

In Results: “Overexpression of Axin1 resulted in a slight reduction in the expression of *chrm2a* (Figure 6—figure supplement 3), indicating that Wnt/β-catenin can work at the level of the M2 receptor.”

In Discussion: “The M2 mAChR is expressed in the zebrafish heart from 30 hpf onwards and by 3dpf the autonomic response is mature (Hsieh and Liao, 2002, Dlugos and Rabin, 2010, Shin et al., 2010). Expression of the M2 mAChR was reduced after inhibiting Wnt/β-catenin signaling (Figure 6—figure supplement 3), suggesting that Wnt/β-catenin signaling effects M2 receptor levels.”